# Seasonal Effects on Health Status and Parasitological Traits of an Invasive Minnow in Iberian Waters

**DOI:** 10.3390/ani14101502

**Published:** 2024-05-18

**Authors:** David Almeida, Juan Diego Alcaraz-Hernández, Alejandra Cruz, Esther Lantero, David H. Fletcher, Emili García-Berthou

**Affiliations:** 1Facultad de Medicina, Universidad San Pablo-CEU, CEU Universities, Urbanización Montepríncipe, 28660 Boadilla del Monte, Spain; alejandra.cruzvarona@ceu.es (A.C.); esther.lanterobringas@ceu.es (E.L.); 2GRECO, Institute of Aquatic Ecology, University of Girona, M. Aurèlia Capmany 69, 17003 Girona, Spain; jdalcaraz@gmail.com (J.D.A.-H.); emili.garcia@udg.edu (E.G.-B.); 3TRAGSATEC Group, 28037 Madrid, Spain; 4UK Centre for Ecology and Hydrology, Environment Centre Wales, Bangor LL57 2UW, UK; dfletcher@ceh.ac.uk

**Keywords:** fluvial ecosystem, life-cycle complexity, Mediterranean stream, non-native fish, parasite abundance, parasite diversity

## Abstract

**Simple Summary:**

Assessments of invasive species are of particular relevance in the Iberian Peninsula, where the endemic fish fauna is highly important for conservation. Parasites of invasive fish can vary substantially because of the seasonal variability in freshwater habitats. The study species was the invasive Languedoc minnow, native to France. Fish organs and parasites were assessed across seasons in a Mediterranean stream, which has a wide range of habitat variability throughout the year. Autumn was the most ‘benign’ season in the study area, with minnows displaying a better health status and a lower parasite burden. This information is relevant to understanding the effects of seasonal variation on parasites of invasive fish. Also, these data could assist environmental managers in controlling the spread of this non-native fish in Mediterranean streams.

**Abstract:**

Biological invasions are of special conservation concern in the Iberian Peninsula and other regions with high levels of endemism. Environmental variability, such as the seasonal fluctuations of Mediterranean streams, is a key factor that affects the spread of aquatic species in novel habitats. Fish parasites have a great potential to reflect such changes in the habitat features of freshwater ecosystems. The aim of this study consisted of seasonally analysing the health status and parasitological traits of non-native fish in Iberian waters. In particular, a strongly invasive population of Languedoc minnow *Phoxinus septimaniae* (leuciscid species native to south-east France) was assessed in Tordera Stream (north-eastern Iberian Peninsula, Mediterranean conditions). Fish were sampled in April, July, and October 2023 by electrofishing. Health status (external/internal organs) was significantly better in autumn (HAI = 28.8) than spring (HAI = 35.6). Life-cycle complexity was higher in spring (LCI = 1.98), whereas parasite abundance and Shannon diversity were significantly lower in autumn (TA = 19.6 and *H’* = 2.15, respectively). In October (more ‘benign’ environmental conditions in Iberian streams), minnows could display elevated foraging activity, with fish increasing their health condition and level of parasite resistance/tolerance. Overall results showed a particular seasonal profile of health and parasite infra-communities that allow this minnow species to thrive under highly fluctuating habitat conditions. This information could help environmental managers to control non-native fish in Mediterranean streams.

## 1. Introduction

The impact of invasive fishes is of particular conservation concern in Mediterranean ecosystems because of the high level of endemism, as well as the numerous anthropogenic threats [1,2]. Predation and competition for food are considered to be the main impacts posed by invasive fishes in the Iberian Peninsula, although the mechanisms of genetic introgression, habitat alteration and disease vector also occur [2]. A key factor affecting the spread of non-native fishes is the environmental variability of novel habitats throughout the year. Particularly in Mediterranean streams, flow and temperature regimes fluctuate a great deal between seasons, from cold-rainy winters to hot-dry summers [3]. Such predictable conditions pose strong abiotic constraints to native communities over an annual cycle, with these populations displaying unique adaptations to thrive (e.g., against a low oxygen concentration in summer, see Gasith and Resh [3] for a comprehensive review). Most of the Iberian Peninsula has a Mediterranean climate. Thus, tropical species may survive after introduction at these higher latitudes in spring or summer (e.g., see the cases of guppy *Poecilia reticulata* and pacu *Piaractus brachypomus* at ≈40° N in Spain [4]). However, many of these species barely overcome the low temperatures during winter, which prevents their establishment and spread across Iberian waters. Similarly, fish introductions from Temperate regions may find ‘mild’ winters in Iberia, whereas they (or their fry) cannot survive during the summer, mainly because of high temperatures (e.g., Danube salmon *Hucho hucho* [4]). Unfortunately and despite this harsh ‘Mediterranean’ context, a wide variety of non-native fishes coming from diverse geographical ranges do survive, establish, and, finally, become invaders [5]. Thus, there is a need for more research to identify particular bio-invasion ‘profiles’ that successfully adapt to such stressors emerging from strong seasonal fluctuations [5]. These data will aid environmental managers in controlling their spread across water courses on the Iberian Peninsula. In particular, many minnow species (taxonomic genus *Phoxinus*) have been introduced into several Mediterranean countries in recent decades [6,7]. For this genus, the onset of reproductive migration is around April, finalizing in May with the first spawns. Age at maturity is usually two years and size of sexual maturity begins at 50 mm. *Phoxinus* species are bottom-feeders, mainly on benthic invertebrates [8].

Among the variety of biological communities that are able to reflect the habitat variability in freshwater ecosystems, fish parasites are regarded as having great potential [9]. Overall, the assessment of parasites and host traits could be used as an appropriate model to provide valuable insights into the effect of co-invaders on native communities (see Lymbery et al. [10] for a seminal review). Such biotic interactions are of great relevance for scientists working on aquatic invasions at the global scale, as >50% of non-native hosts are freshwater fishes [10]. Indeed, these authors also found that virulence was usually greater in native hosts than in the alien species with which the parasite had been introduced. A potential explanation for this finding would be related to a weak defense response due to a lack of co-evolutionary history. On a broader geographical scale (i.e., beyond the Iberian Peninsula), the ‘explosive’ range extension of the Ponto-Caspian round goby *Neogobius melanostomus* in central Europe is largely explained by the co-introduction of nematod and acantocephalan species. Particularly for the Rhine River, these parasites have a strong impact on native fishes which favors the goby invasion within this fluvial system [11]. In Australian fresh waters, poeciliid fishes can inadvertently ‘use’ parasites as biological ‘weapons’ against native species, as they have not had time enough to evolve an equilibrium relationship [12]. In South America, the parasitic copepod *Lernaea cyprinacea* was co-introduced along with the common carp *Cyprinus carpio*. More recently, this parasite species has been found in Colombia, Uruguay, Brazil, Chile, and Argentina, with native hosts displaying higher infection rates [13]. In North America, native amphibians are threatened by Asian *Gyrodactylus* (a monogenean genus of flatworms) as a result of co-introduction with the invasive weather loach *Misgurnus anguillicaudatus* [14]. Africa is another continent where these co-invasions occur, with the cestode species *Atractolytocestus huronensis* and *Bothriocephalus acheilognathi* spreading after the introduction of common carp [15]. Moreover, non-native fish are able to act as competent ‘amplifiers’ for endemic parasites, increasing their transmission to native fauna (see the case of Australian nematod *Contracaecum bancrofti* and its effect on invasive weather loach and common carp in Shamsi et al. [16]). Unfortunately, few studies have provided parasitological information on invasive fishes in Iberian fresh waters (but see Benejam et al. [17] and Cruz et al. [18]), which reveals a deep knowledge gap for this eco-region.

In combination with data on parasitological traits, the assessment of fish health can help to complete a more realistic ‘picture’ of interactions between host physiological/immune responses and the structure of their infra-communities (e.g., species displaying direct or indirect life-cycles). Specifically for invasive fish, health status and parasite burden/diversity are good correlates to their colonization capacity (see examples for an American centrarchid in Rubenson and Olden [19], and for the present study minnow species in Cruz et al. [18]). By adding a seasonal ‘perspective’ to these assessments, synergistic effects may be more easily unveiled between biological (i.e., parasitological/physiological, in this case) parameters and environmental-specific responses under a broad temporal variability in habitat conditions. In this regard, Dias and Tavares-Dias [20] found a clear change in parasite communities of fish assemblages between ‘rainy’ (e.g., higher abundance of the copepod *Perulernaea gamitanae*) and ‘dry’ (e.g., higher abundance of the ciliate *Ichthyophthirius multifiliis*) seasons in the Amazon River system (Brazil). These findings were mainly explained by the strong hydro-dynamic variation, together with the availability of infectious stages in the environment. In North America, the coho salmon *Oncorhynchus kisutch* showed the greatest infestation rates of diplostomid parasites (black spot disease) during summer. This was explained as a consequence of ‘thermal stress’ on fish health, i.e., with higher water temperatures resulting in decreased immune resistance and increased susceptibility to parasites [21]. Fresh waters often show profound changes over the annual cycle in northern Europe, where the winter period is proven to be an important transmission window for parasites, such as the cestode genus *Crepidostomum* in the intestinal tract of Arctic charr *Salvelinus alpinus* and brown trout *Salmo trutta*. Both salmonid species prey substantially on amphipods (intermediate host) during the cold season [22]. Yet to the best of the authors’ knowledge, no previous publication exists focused on Iberian waters where combined data of parasites and fish health, along with their seasonal variation, have been presented. Consequently, the aim of the present study was to analyze the seasonal effects on health status and parasitological traits of an invasive fish species on the Iberian Peninsula. Specifically, an external/internal health index, parasite abundance, parasite diversity, and an index of life-cycle complexity were compared across seasons.

## 2. Materials and Methods

### 2.1. Study Area and Fish Population

Fish were collected from Tordera Stream (NE Iberian Peninsula). The non-native study species was the Languedoc minnow *Phoxinus septimaniae* (hereafter, only ‘minnow’), which is a small-bodied leuciscid fish native to south-east France (an area with river basins under a clear influence of the Temperate climate). This particular stream system and minnow population have previously been well-described in terms of climatic, geo-morphological, limnological, and ichthyological features (see Cruz et al. [18] for detailed text and a comprehensive list of references therein).

Briefly, Tordera Stream is a watercourse around 65 km in length, draining into the Mediterranean Sea (Figure 1), which displays a typical Mediterranean flow regime (i.e., winter floods and summer droughts). The study species of minnow has so far only been detected in NE Spain and it was introduced into Tordera Stream around the year 2000 [23]. The purpose for this introduction was probably related to stocked populations of brown trout *Salmo trutta* (i.e., minnow as a ‘forage’ fish to maintain sport fishing in the upper reach). The minnow invasion was very rapid along Tordera Stream and this fish species is currently the most abundant in the middle reach of this water course, with spawning occurring mainly around mid-May. In terms of ecological impact, minnows likely out-compete native fish assemblages (e.g., Mediterranean barbel *Barbus meridionalis* and Catalan chub *Squalius laietanus*) for food and habitat. Large numbers of this species can be captured in the Tordera Stream with no restriction (i.e., they have non-native status), with this fish extraction having virtually no effect on subsequent data acquisition (i.e., the samples are a small fraction of the overall stream population; see sampling sizes below).

### 2.2. Field Sampling

Fish were seasonally collected during one week in the middle of the month, in April (spring), July (summer), and October (autumn) 2023. These sampling periods were just before and after the breeding season of the minnow (May, see above), which avoids any effect of this physiological (i.e., reproductive) status on data. Fish sampling was not carried out in winter, to avoid disturbing trout spawning. The year 2023 is considered to have been hydrologically ‘average’ in the study area [24]. Thus, the effects of particularly dry or wet years on health status and parasitological traits were avoided, which allows the data to be considered representative for this fish species and their parasite infra-communities (see Bush et al. [25] for term definitions) in Tordera Stream.

According to the European legislation (CEN/ISO Standards), field surveys followed a consistent electrofishing protocol to allow the acquisition of representative and comparable minnow samples from the study stream (see the comprehensive design in Merciai et al. [26]). Briefly, three sampling sites (100 m river stretch per site) were surveyed along the main middle reach of Tordera Stream (≈30 km). To minimize spatial dependence, these sites were evenly distributed along this stream reach (>10 km of separation, Figure 1) and they were similar in terms of environmental features (e.g., riparian vegetation, land-use).

The present study was conducted in agreement with the relevant legislation of Europe and Spain, by means of a fishing permit granted by the regional authorities of Catalonia: SF/0193/2023. Only non-native species (including minnow) were collected. Thus, no adverse effects were caused to the wildlife/habitat in the sampling sites, with special care taken of native fishes. After electrofishing was concluded at each site, and in accordance with current regulations, non-native fishes were euthanized in the field (see below for euthanasia protocol), except for minnows, which were first transported back to the laboratory in cooled and oxygenated water (within 1 h of capture). All minnows from one particular site (see sampling sizes below) were processed in the afternoon of the day of capture. Thus, fish stress was minimized and disturbances in their infra-communities (e.g., specimen loss) were reduced (i.e., data on parasites are more naturally representative/realistic).

### 2.3. Laboratory Procedures

Morphological and parasitological examinations followed laboratory procedures well-described in Cruz et al. [18] and Almeida et al. [27], as well as references therein. On arrival at the laboratory, fish were stored in 50 L aquaria after a 30 min period of acclimatization. According to animal use and care regulations, each individual minnow was euthanized before examination. The euthanasia protocol consisted of an overdose of clove oil, followed by severance of the spinal cord (adapted from Chapman et al. [9]). After euthanasia, specimens were measured for Standard Length (SL, to the nearest 1 mm). The largest specimens were selected for assessment because health and parasitological patterns are clearer for this small-bodied fish species within the study stream system. In total, 300 mature individuals (33–34 ind. × 3 sites × 3 seasons) were examined, with size ranges being (mm SL) 55–74 in spring, 55–78 in summer, and 55–76 in autumn. No significant differences were found for fish size between seasons (ANOVA: *F*_2, 297_ = 2.25, *p* = 0.107), with means of 60.1 mm SL (SE = 0.78) in spring, 62.4 mm SL (0.86) in summer, and 60.6 mm SL (0.82) in autumn. After body length measurements, fish were dissected and sex determined. Eviscerated body mass (eBM) was obtained by using an electronic balance (±0.1 mg).

A Health Assessment Index (‘HAI’, hereafter) was computed, as per Adams et al. [28]. Specifically, HAI consists of examining a variety of external/internal organs: skin, eyes, gills, alimentary tract, heart, liver, spleen, and gonads. Abnormalities, color, size, and condition are assessed for each organ, which is assigned a score ranging from 0 (healthy) to 30 (unhealthy) points (see Almeida et al. [27] for the particular algorithm). Theoretically, the maximum value for HAI would be 240 (i.e., 30 points × 8 organs) per specimen. However, no fish exceeded a total score of 100 points in this study. Additionally, HAI values are also used to complement data on parasitological traits (see details below), given that this index can be considered as a proxy of ‘tolerance’, according to Blanchet et al. [29].

During examinations to compute HAI, both ecto- and endo-parasites were also identified and counted per fish individual. In the present study, >30 fish specimens per site were examined (see sampling sizes above) for parasites, which is a sufficient number to achieve a good representation of infra-communities for this minnow species [18], as well as other fishes (see an example for pipefish and gobies from a Mediterranean coastal lagoon in Almeida et al. [27]). Parasites were identified under dissecting (40×) and light (1000×) microscopes, using specific keys [30,31,32]. Genus was the lowest practical taxonomic level reached for two parasite taxa (Table 1). According to Poulin [33] and Chapman et al. [9], taxonomic genera provide enough information to assess the ecological roles within the fish–parasite relationship (e.g., definitive/intermediate host, monoxenous/heteroxenous parasites).

### 2.4. Data Analyses

After preliminary Linear Models, no difference was found between males and females on the examined traits (*p*-values > 0.05). Therefore, the categorical factor ‘sex’ was not included in subsequent data analyses, which simplified biological interpretations. Also, data were pooled because the effect of ‘sampling site’ was not significant, after previous Generalized Linear Mixed Models (GLMMs, with ‘site’ as the random factor).

To explore temporal patterns, quantitative descriptors were used to properly analyze the variation in parasitological traits across seasons. Three parameters of parasite infra-communities were calculated for each fish individual (adapted from Bush et al. [25]): ‘Total Abundance’ (TA, total number of parasite individuals detected from the examined organs), ‘Diversity’ (Shannon index *H’*, calculated by using log_2_) and a ‘Life-cycle Complexity Index’ (LCI, see a detailed explanation and formula in Almeida et al. [27]). The number of different hosts ranged from one to three species for the parasite taxa found in the present study and consequently, the descriptor LCI (weighted by parasite abundances) also varied within this range. This parameter can be regarded as a proxy of local ‘biodiversity’, with high values indicating an elevated abundance of heteroxenous parasites that need several ecologically diverse hosts (e.g., zooplankton, mollusks, fish, birds, or even aquatic mammals) to complete their life-cycles.

One-way ANOVA was used to confirm that similar fish sizes (SL) were selected between seasons (see results above). Despite no differences being found in fish length, the effect of body size/mass was controlled in subsequent comparisons by using Analysis of Covariance (ANCOVA). This technique allows analysis of the potential effect of mass on response variables, as well as different statistical interactions compared to ANOVA. In particular, ANCOVA (covariate: eBM) was used to test for significant differences between seasons for the response variables HAI and the three parasite descriptors (TA, *H’*, and LCI). In case of statistical differences, a post-hoc Tukey–Kramer Honestly Significant Difference (HSD) test was used to perform pair-wise comparisons between seasons.

Data were transformed by using log_10_ (*x* + 1). Residual plots and Levene’s tests were used to verify assumptions of normality of residuals and homogeneity of variances, respectively. Statistical analyses were performed with R v.3.6.3 [34]. Values reported in results are means ± SE/SD.

## 3. Results

After controlling for body mass, differences among seasons were found for HAI (ANCOVA: *F*_2, 296_ = 5.27, *p* = 0.006), which was significantly lower (i.e., better fish health) in autumn (28.8 ± 1.54) than in spring (35.6 ± 1.58) (Tukey’s HSD test, *p* < 0.05). HAI for minnows showed an intermediate value (32.0 ± 1.61) in summer, compared to autumn and spring (Tukey’s HSD test, *p* > 0.05).

A total of 15 parasite taxa were identified throughout the three seasons (Table 1).

The nematode *Raphidascaris acus* was not detected in spring or autumn, whereas the digenean *Hysteromorpha triloba* was not found in summer (Table 1). A wide seasonal variation was observed for particular parasite taxa. The crustacean *Ergasilus sieboldi* was clearly more abundant in autumn. However, the most common seasonal pattern consisted of a higher parasite abundance in spring-summer, relative to autumn. Thus, abundances of the two digenean genera *Posthodiplostomum* and *Diplostomum* were around seven and five times higher in April than October, respectively. The monogenean species *Dactylogyrus phoxini*, *Gyrodactylus macronychus*, *G. phoxini*, and the ciliate genus *Trichodina* were much more abundant in July (Table 1). Consequently, the three parasite descriptors showed a strong variation across seasons (Table 2), according to the features of these infra-communities (i.e., taxonomic proportions shown in Table 1). Specifically, TA and *H’* were significantly lower in autumn, whereas LCI showed the highest values in spring (Table 2).

## 4. Discussion

Like many other European small-bodied leuciscid fishes, the Languedoc minnow displays partial potamodromous behavior [8], with a fraction of the population migrating from wider main-stems to the narrower head-waters/tributaries for spawning. Such a strategy (i.e., a seasonal migration for reproduction within a particular fluvial system) has also been observed after minnow introductions in Iberian water courses, including the Tordera Stream [4]. The present results on health status may be understood according to the Dynamic Energy Budget (DEB) theory, which offers a systematic way to relate processes at different organizational levels, from molecules to ecosystems [35]. This is a perspective on theoretical ecology whose starting point is homeostasis and bioenergetics, linking metabolic/physiological mechanisms within individual organisms to population responses [35]. In April, just before the breeding period (mid-May), this fish species is allocating a high level of its energy budget to locomotive activity at the expense of a detrimental effect to its health condition (see a comprehensive review on factors affecting fish migration in Wanjari et al. [36]). This is a potential explanation for the observed higher minnow HAI (i.e., worse health condition) in spring. In July, and after the intense reproductive effort, minnows are able to display a slight recovery of health, in spite of the typical features of Mediterranean summers (i.e., less flow, higher temperature, and lower oxygen level). In October, minnows take advantage of the more ‘benign’ environmental conditions throughout autumn (i.e., moderate levels of water current, temperature, and oxygen) to display elevated levels of foraging activity (D. Almeida, pers. observ. from snorkel surveys). Consequently, fish can increase reserves before the ‘cold season’, along with their health level (i.e., showing the lowest HAI), which contributes to improving the over-wintering survival rate [37].

These results on minnow health could also be related to the seasonal variation in parasitological traits. To understand this, it is useful to highlight that schooling behavior is commonly displayed by the taxonomic genus *Phoxinus* [38]. In April, migratory events promote closer physical contact among minnows within the same shoal (D. Almeida, pers. observ.). This facilitates direct transmissions of monoxenous parasites, with the abundance being high during this month. Following the same pattern as the HAI results, parasite abundance decreased from spring-summer to autumn, corresponding with a shift to a lower level of social aggregation [38]. Other parasite community traits (e.g., burden and diversity) can also reflect the physiological status of hosts, as these biotic relationships (i.e., parasite *vs.* fish hosts) would change depending on the allocation of energy resources, in the context of DEB theory (see above).

Mechanistically, minnows may more easily develop parasitic diseases due to a lower investment of their energy budget for the immune response under spring (migration stress) and summer (thermal stress) conditions [39]. Moreover, hosts can protect themselves against parasites by limiting the damages caused by these infra-communities (i.e., ‘tolerance’) or by actively reducing parasite burden (i.e., ‘resistance’) [29]. Both physiological responses (i.e., tolerance and resistance) were higher under autumn conditions, with minnows displaying a better health index and lower parasite abundance, respectively. Nevertheless, presence of parasites is the most common and natural status in fish. Indeed, most parasites are not highly pathogenic and fish do not develop serious diseases (i.e., a history of co-evolution). Therefore, a lower parasite abundance does not always mean a ‘healthier’ fish, and this was a good reason to use a specific health index along with parasitological traits. Regarding parasite diversity, the decrease in the infra-community abundances was not proportional for each taxa, from July to October, with some particular species being favored over others in the environmental conditions of autumn (see a similar example for pumpkinseed sunfish *Lepomis gibbosus* in Chapman et al. [9]). More specifically, only three (monoxenous) parasite taxa accounted for ≈70% in total abundance during autumn (*Trichodina*, *Dactylogyrus*, and *Ergasilus*), consequently causing a significant decrease in diversity. However, healthier fish can also ‘bear’ a higher parasite diversity, which is commonly found in fresh waters [9]. Additionally, taxa displaying complex life-cycles (i.e., heteroxenous) showed a proportionally higher abundance in spring, according to the LCI result. The most representative examples were the Digenea flukes of genera *Diplostomum* and *Posthodiplostomum*. To complete their life-cycle, these parasite taxa need three different host species: freshwater snails, fishes, and birds. The so-called metacercaria (the last larval stage) often infects the eyes or skin of fish (intermediate host) and alters its behavior to be better detected by piscivorous birds (e.g., herons or gulls as definitive host species), where the parasites then reproduce. Since these parasite species require ‘active’ hosts interacting with each other, the present results demonstrate that fish infra-communities can rapidly change, after just a month of ‘high level’ biotic interactions (April, in this case). Consequently, abundance and richness of heteroxenous parasites can be used as a surrogate of biodiversity and ecological complexity at the local scale, responding within relatively short time-frames (i.e., only one season-period). In terms of co-invasions, introduced parasites displaying such indirect cycles are not expected to establish in novel areas at the same level as species with simple/direct life-cycles. However, Lymbery et al. [10] found that a substantial proportion (35–45%) of co-introductions were of heteroxenous parasites, i.e., with an indirect life-cycle. This is puzzling, given that several alternative hosts (apart from the invasive fish) would be necessary to complete their life-cycles in the new habitats. As specific examples at a broader geographical scale, the nematode *Spirocamallanus istiblenni* is an intestinal parasite, which is currently spreading across the Hawaiian archipelago after the introduction of the bluestripe snapper *Lutjanus kasmira* for the benefit of fisheries. Unfortunately, this parasite has been able to complete its cycle by using native copepods as intermediate hosts [40]. In South Africa, the Asian cestode *Schyzocotyle acheilognathi* co-invaded the region on common carp, goldfish *Carassius auratus*, and most likely the grass carp *Ctenopharyngodon idella* (introduced as biological control for invasive aquatic plants). Currently, this intestinal parasite has been found in the African sharptooth catfish *Clarias gariepinus*, which means that the invasive cestode can complete its entire life-cycle in native species (copepods and fishes) [41]. Such findings demonstrate the elevated adaptiveness of parasites to ‘use’ novel intermediate and definitive hosts within the introduced ranges.

To the best of the authors’ knowledge, no study has reported data on parasites for this particular fish species, *Phoxinus septimaniae*, in its native range (France). However, the closest information on parasites does exist for its ‘sister’ species *Phoxinus phoxinus*, which usually shows a higher parasite diversity in the native range (>30 taxa) [42,43,44]. More specifically, Pettersen et al. [44] observed that minnow parasites were similar between introduced and native areas in Norway (although this study only focused on the ecto-parasitic genus *Gyrodactylus*). To test for potential limitations of the present study, visual examinations alone may result in underestimation of parasite detection/abundance, as observed by Shamsi et al. [45]. In the present study, two parasite species were not detected in some seasons (the digenean *Hysteromorpha triloba* and the nematod *Raphidascaris acus*), in spite of their life-spans being longer than a single season (three months) [46]. Maybe advanced techniques, such as digestion or incubation methods [47,48], could be applied to yield more accurate data for these parasite species. In addition, future research should focus on a broader size range. In particular, the assessment of health and parasites in juvenile minnows could shed light on population recruitment along Tordera Stream (see an example for two leuciscid fishes in Longshaw et al. [49]). In application, the information generated by this study suggests that, in order to be most effective, control measure efforts should be concentrated at times when minnows are more easily detectable (‘active’) and vulnerable (i.e., in worse health, during migration and reproduction months).

## 5. Conclusions

This work shows that invasive minnows appear to display a particular seasonal profile of health status and parasitological traits that allow this ‘Temperate’ fish species to persist and thrive under highly fluctuating environmental conditions, such as those found in Mediterranean streams [3]. Unfortunately, this abundant population (Tordera Stream) could act as a ‘source’ facilitating subsequent invasions in other water courses across the Iberian Peninsula. Nonetheless, these findings could also assist environmental managers in identifying vulnerable areas (according to habitat conditions), where monitoring programs can be established for early detection. Additionally, overall data on parasites would be useful to establish a community baseline and focal species for the design of particular fish assessments applied to Mediterranean streams at a broader scale. Such information will surely contribute towards reducing the spread of non-native fish populations in Iberian waters.

## Figures and Tables

**Figure 1 animals-14-01502-f001:**
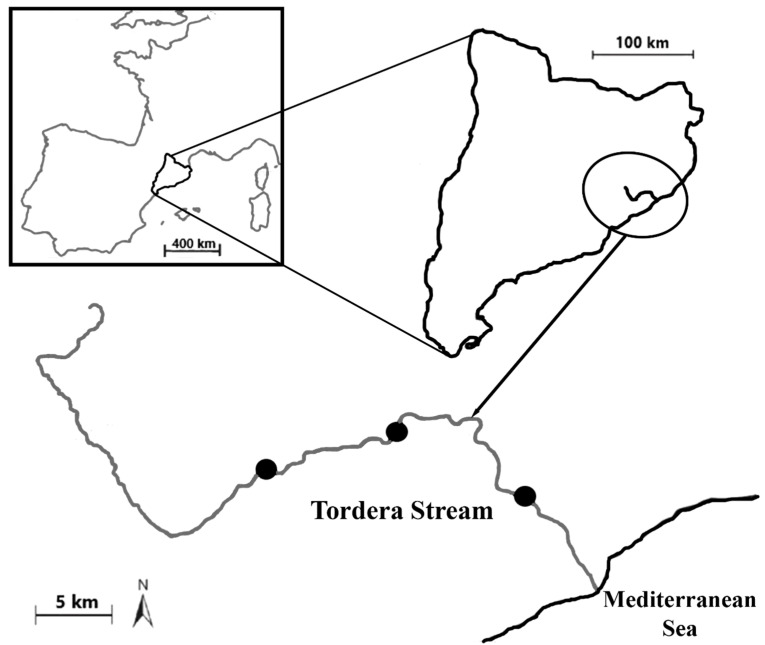
Map of the study area showing the geographic location for Tordera Stream (Catalonia Region, NE Spain). Black circles: sampling sites.

**Table 1 animals-14-01502-t001:** Parasite mean abundances (±SD) in Languedoc minnow *Phoxinus septimaniae* from Tordera Stream across the three sampling seasons. Dashes indicate undetected parasites on examined minnow individuals (*n* = 82 for spring, 72 for summer, and 50 for autumn).

Parasite (Taxon)	Parasite (Genus/Species)	Spring	Summer	Autumn
Ciliophora	*Trichodina* sp.	2.7 ± 2.01	5.4 ± 2.68	4.0 ± 2.77
Cestoda	*Proteocephalus exiguus*	0.6 ± 0.58	0.8 ± 0.40	0.2 ± 0.42
Monogenea	*Dactylogyrus phoxini*	5.1 ± 3.08	7.2 ± 3.82	5.1 ± 3.11
	*Paradiplozoon homoion*	1.2 ± 1.13	0.6 ± 1.05	1.3 ± 1.17
	*Gyrodactylus macronychus*	0.9 ± 1.15	3.4 ± 1.74	0.2 ± 1.04
	*Gyrodactylus phoxini*	1.4 ± 1.27	4.0 ± 1.56	0.1 ± 0.99
Digenea	*Allocreadium transversale*	0.8 ± 0.87	0.3 ± 0.80	0.5 ± 0.79
	*Diplostomum* sp.	4.9 ± 2.04	1.1 ± 1.84	1.0 ± 1.87
	*Hysteromorpha triloba*	0.8 ± 0.85	–	0.7 ± 0.99
	*Ichthyocotylurus platycephalus*	0.1 ± 0.61	0.5 ± 0.50	0.6 ± 0.49
	*Posthodiplostomum cuticola*	7.3 ± 2.25	2.2 ± 2.50	1.0 ± 2.34
	*Tylodelphys clavata*	1.0 ± 1.88	1.4 ± 2.06	0.4 ± 1.45
Nematoda	*Camallanus lacustris*	0.5 ± 1.33	0.3 ± 0.71	0.3 ± 0.47
	*Raphidascaris acus*	–	0.1 ± 0.23	–
Crustacea	*Ergasilus sieboldi*	0.2 ± 0.93	1.2 ± 1.16	4.2 ± 1.74

**Table 2 animals-14-01502-t002:** Parasitological traits (TA, *H’*, and LCI) for Languedoc minnow *Phoxinus septimaniae* from Tordera Stream. Results are (adjusted) means ± SE, reported per season. *F*-ratios, degrees of freedom, and significance levels (*p*-values) are presented, after ANCOVA (covariate: eviscerated mass, see ‘Data Analyses’ for details). Significant results are highlighted in bold. Lower-case superscript letters indicate significant differences between seasons (Tukey’s HSD test, *p* < 0.05).

	Spring	Summer	Autumn	*F* _2, 200_	*p*-Value
TA	27.5 ± 2.19	28.5 ± 2.31	^a^ 19.6 ± 2.37	3.50	**0.032**
*H’*	2.38 ± 0.040	2.43 ± 0.042	^a^ 2.15 ± 0.047	4.62	**0.011**
LCI	^a^ 1.98 ± 0.096	1.54 ± 0.094	1.53 ± 0.102	3.73	**0.026**

## Data Availability

Data are available from the authors upon reasonable request.

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
