# Peer review of "Seasonal Effects on Health Status and Parasitological Traits of an Invasive Minnow in Iberian Waters"

_animals, 2024, doi:10.3390/ani14101502_

Round 1

Reviewer 1 Report

Comments and Suggestions for Authors

Health status and parasitological traits of the invasive minnow Phoxinus septimaniae were seasonally investigated in Tordera Stream. A seasonal profile of fish health and parasite infection were found, which was responsible for the season changes. But these findings were not relevant to invasive process of the fish, and did not support some viewpoints of the authors’. Specific comments are presented as follows.

Simple summary:

Line 23-24: Parasitism is a common phenomenon. Of course, parasites are usually detected in fish from the new or introduced habitat. Usually, composition and diversity of parasite community are affected when the fish hosts were introduced to new environment, but not “Parasites can modulate the invasion process in freshwater fish”. Thus the incorrected viewpoint may lead to the bad explanation on the results.

Line 28-31: A better health status and a lower parasite burden in autumn may be normal, but irrelevant to invasion process.

Abstract

Line 44-47: Detailed or specific results should be presented here, such as the specific health status and parasite infection.

Line 47-57: Too many inferences and explanations.

Line 49: Mean abundance may is a better parameter than the prevalence because the prevalence of all the parasites is meaningless and did not indicate the infection level.

1. Introduction

Line 75-76: The reference (Britton, 2013) did not support this point! Please check. As mentioned above, the viewpoint may be wrong.

Line 77-83: The statement on hypotheses of parasites of the introduced fish is not related to the present study because no native and exotic parasites were differentiated and no lost parasites were determined.

Line 108-109: More biological habits should be provided here, such as migration time of reproduction, age/body length of the matured minnow, feeding habits, which are very important to understand the present results.

2. Materials and methods

Line 173: It is better to cite the reference, Bush et al., 1997, where there are lots of term definitions on parasite infra-community.

Line 207-208: Are these fish matured?

Line 242: As mentioned above, the overall prevalence is not a good descriptor to measure the infection level of all the parasites, but mean abundance.

3. Results

Line 283: Replace parasite loads with parasite mean abundance, and add the SD in the Table 1.

Line 288-294: It is improper to compare mean abundance among the different parasite species in the same season because of the huge difference in size among different species. Perhaps it is better to summarize patterns of the seasonal change.

4. Discussions

Line 307-329: Total length of the fish caught in the present study is about 60 mm. Does the examined fish reach to sex maturation? If not, it is unsuitable to discuss the relationship between spawn migration/energy allocation and and health status.

Line 341-367: It is common and normal to find parasites from fish. Higher diversity is usually detected in healthier fish. Furthermore, most parasites are not pathogenic, and do not cause diseases. Therefore, lower parasite abundance does not mean healthier.

Line 368-382: No any result on lost parasites was provided in this study compared with the native parasites in the native range. So it is unnecessary to discuss the hypotheses on parasites of the invasive fish.

Comments on the Quality of English Language

The manuscript was well written in English.

Reviewer 2 Report

Comments and Suggestions for Authors

This is potentially an important topic; however, some aspects need improvement.

Missing from the manuscript: please include a list of parasites reported from this fish in France where the fish is endemic and in the study area where it is considered introduced.

The journal Animals has an international audience and is not limited to a specific region; therefore, the outcomes of the study should go beyond just the study region. However, both in the Introduction and the Discussion, authors only talk about the significance of the study in the Iberian region and do not include the rest of the world, which is in contrast with the "one health" concept they discuss early in the manuscript.

Another major comment is that despite talking about "one health," the manuscript basically is about the seasonal variation of a number of parasites in a fish, and discussion of the findings is mainly speculative.

Line 72: should it be ‘competition’?

Line 76: Authors may also be interested in reading this paper: Proceedings of the Royal Society of Queensland, 1998, 107, 109-113.

Lines 77 onward: I suggest reading and reflecting on this paper: International Journal for Parasitology: Parasites and Wildlife, 2014, 3, 171-177. Particularly, using the term ‘enemy’ does not seem right.

Lines 83 and 84: Authors say there hasn’t been much study in their study region. Therefore, it is important to briefly explain what we know in this regard in other parts of the world. This is literature review and background of the study, and authors should do a more in-depth review of the literature. They may also be interested in reading this paper where authors showed an introduced species, carp, picked up native parasite, Contracaecum bancrofti, and became an amplifier of the parasite among other native fish: Journal of Helminthology 2019 Vol. 93 Issue 5 Pages 574-579.

Lines 124 to 126: Again, authors say there is no other study in their study region, but what about other parts of the world? Has anyone done that? And what did they find? Please briefly explain.

Lines 130 to 131: Sounds like methodology, and I suggest moving it to Materials and Methods.

Lines 196 to 233: I understand that the study relies solely on necropsy and visual examination methods for parasite detection, which may result in underestimation of parasite prevalence. Employing more advanced techniques such as digestion or incubation methods (see Canadian Journal of Fisheries and Aquatic Sciences 1986 Vol. 43 Issue 7 Pages 1312-1317; International Journal of Food Microbiology 2016, 227, 13-16) could yield more accurate results. It's possible that you may have missed some parasites in your fish. For example, in an article published in Marine and Freshwater Research in 2023 (Volume 74, Issue 12, Pages 1095-1101), the authors mention that no parasites were found when only visual examination was employed. Therefore, this limitation must be noted in the study.

In the result section, authors mention some internal parasites such as Raphidascaris has not been found in all seasons. Are you suggesting that their lifespan is too short then? Because this is against what previously other researchers stated.

Comments on the Quality of English Language

There are minor typos that can be corrected if authors carefully review the manuscript.

Round 2

Reviewer 1 Report

Comments and Suggestions for Authors The manuscript has been  completely corrected according to the comments, and now it is greatly improved.

Reviewer 2 Report

Comments and Suggestions for Authors

Authors addressed my comments.